# Non-Invasive Real-Time Monitoring of Bacterial Activity by Non-Contact Impedance Spectroscopy for Off-the-Shelf Labware

**DOI:** 10.3390/s25082427

**Published:** 2025-04-11

**Authors:** Carsten Thirstrup, Ole Stender Nielsen, Mikael Lassen, Thomas Emil Andersen, Hüsnü Aslan

**Affiliations:** 1Danish Fundamental Metrology Kogle Allé 5, DK-2970 Hørsholm, Denmark; cth@dfm.dk (C.T.); osn@dfm.dk (O.S.N.); ml@dfm.dk (M.L.); 2Department of Clinical Microbiology, Odense University Hospital, J. B. Winsløws Vej 21, 2nd Floor, DK-5000 Odense, Denmark; 3Research Unit of Clinical Microbiology, University of Southern Denmark, J. B. Winsløws Vej 21, 1st Floor, DK-5000 Odense, Denmark

**Keywords:** non-contact, electrodes, impedance analysis, Raman spectroscopy, machine learning, labware

## Abstract

Monitoring bacterial activity is essential for numerous scientific and industrial applications. However, current benchmark measurements, i.e., optical density (OD), exhibit a limited dynamic range and require transparent or translucent media. Conventional impedance spectroscopy involves direct electrode contact with the bacterial medium or biofilm, potentially perturbing the sample environment and compromising measurement fidelity. Moreover, many real-time methods rely on costly, specialized labware that limits scalability and versatility. Here, we introduce a non-contact impedance spectroscopy (NCIS) technique with customizable electrodes for off-the-shelf labware and show that the data collected from a KCl solution series agree well with the simplest electrolytic conductivity cell model solution, demonstrating the accuracy and simplicity of NCIS. As an example of bacterial activity monitoring, NCIS was performed in glass laboratory bottles and 24-well plates in which *Staphylococcus epidermidis* and *Escherichia coli* cultures were inoculated into Brain Heart Infusion media, maintained at 37 °C. Comparative OD measurements acquired intermittently from the same media exhibited a strong correlation between NCIS and OD data, confirming reliability and reproducibility. The bacterial culture was verified by Raman spectroscopy assisted by machine learning. NCIS eliminates the risks of contamination and sample alteration, minimizing costs and operational complexity and providing a scalable, versatile solution for biological and chemical research.

## 1. Introduction

Impedance spectroscopy is a well-established and extensively used method for analysis of solid materials, liquids and interfaces [1]. Impedance-based sensors have proven to be useful for identifying, characterizing and studying the growth of cells, bacteria, biofilms and other micro-organisms, and they are used in many areas of application including industrial processes [2,3] environmental monitoring [4], fermentation [5], and in the food industry [6,7] and healthcare [8,9]. Impedance-based sensors are attractive because they are fast, non-destructive, label-free, enable real-time analysis and do not require transparent samples [2,10,11].

Since the 1990s, impedance spectroscopy has been used for bacterial cell detection and monitoring [12]. The method is less time-consuming, labour-intensive and costly compared to commonly used methods such as biochemical phenotype methods, molecular hybridization methods and amplification methods [13]. Compared to analytical techniques such as chromatography or spectroscopy, impedance spectroscopic methods are simpler and easier to miniaturize [14].

An impedance-based sensor normally comprises two metallic electrodes, such as interdigitated electrodes, or a microarray of electrodes, which are immersed into the medium to be measured. For monitoring the growth of micro-organisms, the sensor detects the physical, chemical and biological phenomena at the interface between the metallic electrodes, the medium and the micro-organisms [10]. However, growth of micro-organisms on surfaces has been demonstrated to be sensitive to the choice of metal electrode material such as stainless steel, indium-tin-oxide, platinum and gold [15]. To avoid electrode fouling and Faradic reactions between the micro-organisms and the electrodes, the electrodes can be coated by an insulation layer and the data recorded as non-contact impedance spectra [16,17].

Yuskina et al. [18] recently reviewed the evolution of contactless conductometry and found a wide number of applications in medical diagnostics, process control and environmental monitoring. For the non-contact method of measuring impedance, there are basically two main coupling schemes, capacitive and inductive. The inductive coupling scheme was developed in the 1950s, primarily for the purpose of measuring salinity in salty waters [19]. It has later been implemented in bacterial growth monitoring, where reports of non-contact inductive methods include a waveguide resonator operating at 16.5–17.3 GHz [20], and a microstrip ring resonator sensor with an operating frequency of 2.5 GHz for detection of growth of pathogenic bacteria [21]. The capacitive coupling scheme has been employed in non-contact electrochemical monitoring for evaluation of biofilms [22], real-time analysis for cytotoxicity testing [23], and for determination of the limit level of aqueous solutions in a container [3].

Common to these electrical non-contact methods is the fact that they cannot be applied to standard labware containers. On the other hand, in electrophoresis, capacitive methods have been developed over the years for dielectric spectroscopy and capacitively-coupled-contactless-conductivity-detection (C4D) [24], where the containers are capillary tubes or glass tubes. Zhang et al. have developed an eight-channel C4D monitoring system for monitoring the growth of bacteria in disposable glass tubes with an excitation frequency of 2.0 MHz and an excitation amplitude of 16 V [4]. The electrode configuration is based on a couple of fixed copper cylinders, where the glass tube is inserted inside the two electrodes. Fixed copper cylinders may apply to standard test tubes but generally cannot be applied to other types of labware which are used in many chemical and biochemical laboratories. Laboratory flasks, bottles, and in particular well plates [25] are extensively used for chemical and microbiological analyses including monitoring of bacterial growth.

The authors of the present paper have developed a sensor concept based on electrode configurations for non-contact impedance spectroscopy (NCIS) that can be used on off-the-shelf labware. Using the electrodes developed for NCIS, it is demonstrated that bacterial growth can be monitored accurately and simply by an inductance–capacitance–resistance (LCR) meter or impedance analyser in laboratory bottles (referred to as lab-bottles) and in 24-well plates. The concept is based on a pair of electrodes which make physical contact with the glass walls of the container, and it can readily be expanded to other geometrical shapes of containers comprising electrically isolated walls. The advantage is that the developed sensor concept can be applied to existing labware without the need for specially designed containers with integrated electrodes. The contact between the electrodes and each container ensures minimum air gap spacing between the electrodes and the outer walls of the container, and the container can be mounted with reproducible positions of the electrodes. Compared to optical measurement techniques such as optical density (OD) and optical reflection (OR), NCIS has the advantage that it provides more details and covers a longer time and a wider dynamic range of the bacterial growth process. As is demonstrated by the data in the present paper, OR primarily monitors initial bacterial growth close to the surface of a lab-bottle, and OD primarily monitors subsequent bacterial growth in the bulk of the lab-bottle, whereas NCIS provides information about both growth processes.

In the following, the term “impedance spectroscopy” will be used both in the context of NCIS and in contact impedance analyses, and it covers data or changes in data of the real part of the impedance and the imaginary part of the impedance as functions of frequency, time or conductivity. The term “impedance spectroscopy” also covers data or changes in data of the imaginary part of the impedance as a function of the real part of the impedance. In Section 2, the design of the electrodes for a lab-bottle and a 24-well plate and the measurement configurations used for NCIS are described. In Section 3, impedance spectroscopy of potassium chloride (KCl) solutions in a lab-bottle and in a well of a 24-well plate are presented for a conductivity range from ultra-pure water (UPW, with a conductivity of 5.5 µS/m) up to 30 S/m. The data are compared with the simplest model of an electrolytic conductivity cell comprising a resistor and a capacitor (RC) in parallel. It is demonstrated that the growth of *Staphylococcus epidermidis* (*S. epidermidis*) in a lab-bottle and in a 24-well plate can be monitored by NCIS, showing a typical sigmoidal growth curve for real and imaginary parts of the change in impedance. Growth data of *S. epidermidis* and *Escherichia coli* (*E. coli*) acquired by NCIS is compared with OR at a wavelength of 800 nm, and it is also compared with the current benchmark, OD at a wavelength of 600 nm (OD_600_). Samples collected by the end of the growth phase are analysed for species identification by Raman spectroscopy assisted by machine learning. Finally, conclusions are presented in Section 4.

## 2. Materials and Methods

Two different designs of electrodes for NCIS are illustrated in Figure 1 and Figure 2, applicable for lab-bottles and multi-well plates, respectively. Figure 1 illustrates electrodes and measurement configurations for a 500 mL glass lab-bottle, which is filled with an electrolyte solution or a bacterial growth medium. The outer diameter of the lab-bottle is (85.6 ± 0.5) mm, and the glass wall thickness is approximately 3 mm. The lab-bottle comprises a top screw-locked circular ring electrode of stainless steel (height, d = 10.0 mm), where the screw ensures tight contact between the top electrode and the outer wall of the lab-bottle. A spring-loaded bottom electrode of copper (ø 19.0 mm) ensures tight contact between the bottom electrode and the outer wall of the lab-bottle. The bottom electrode is surrounded by a guard to reduce the effects of fringe fields. The vertical distance between the top electrode and bottom electrode (h = 82.3 mm) is defined by a partly cylindrically shaped bottle mount as illustrated in Figure 1a–c, with an inner diameter slightly larger than the outer diameter of the lab-bottle. The circular ring electrode is loosened and tightened when unloading and loading the lab-bottle in the bottle mount.

Three different configurations of NCIS measurements as reported in the present paper are illustrated in Figure 1 for a lab-bottle with (Figure 1b) a photograph of the electrodes designed for NCIS and employed for the lab-bottle, (Figure 1c) a photograph of the electrodes attached to the lab-bottle mounted in the bottle mount, (Figure 1a) the lab-bottle equipped with a closed lid, (Figure 1e) the lab-bottle equipped with a pair of polytetrafluoroethylene (PTFE) tubes (3.2 mm inner diam.) immersed into a solution in the lab bottle. The configuration in Figure 1e is inserted in either (Figure 1d) a closed-flow-loop system comprising a liquid control system [26], an electrolytic conductivity coaxial primary cell developed at DFM [27] and a pump circulating the solution, or (Figure 1f) a closed-flow-loop system comprising a pump (Gambro/Premotec, Lund, Sweden) with adjustable flow rate and a 24 VDC three-way solenoid valve with a PTFE diaphragm (Bio-Chem Fluidics, Boonton, NJ, USA). The three-way valve is activated when collecting a sample of the solution for on-line or off-line analyses. For the measurements involving sample collection as depicted in Figure 1f, the lab-bottle is sufficiently filled with the solution ensuring that a change in the liquid level after collecting a sample does not noticeably affect the measured impedance data. It has been verified that this condition is fulfilled provided that the liquid level of the solution as used is more than 6 mm from the top of the upper electrode. In all configurations in Figure 1, a PTFE-coated (ø8.1 × 27.4) mm magnet is immersed in the solution to facilitate stirring at 400 rpm using an RCT Basic stirrer (IKA-Werke, Staufen, Germany).

The impedance data are recorded by a Keysight E4980 LCR meter (Keysight Technologies, Santa Rosa, CA, USA) in the frequency range from 10 kHz to 2 MHz and with a voltage amplitude of 1 V. Optical reflection (OR) from an area in the vicinity of the inner lab-bottle wall is measured using a reflection probe based on an optical fibre bundle (RP 20 from Thorlabs, Newton, MA, USA), a glass lens with an effective focal length of 5.5 mm (Edmund Optics, Barrington, NJ, USA), and an SLS201L/M halogen lamp (Thorlabs, Newton, MA, USA) for illumination. Using an Ocean Insight miniature spectrometer model Ocean SR (Ocean Optics, Duiven, The Netherlands), reflected optical spectra are recorded simultaneously with the impedance spectra. The lab-bottle and bottle mount are mounted in an aluminium box, which is kept in a temperature-controlled air-bath with a setpoint just below the target temperature. The box comprises heater elements for temperature control and shields for electromagnetic interference. A calibrated Pt100 sensor (AccuMac Technology, Gilbert, AZ, USA) measured by a Fluke 1502A thermometer (Fluke Corporation, Everett, WA, USA) readout monitors the temperature.

Figure 2 illustrates an electrode configuration for NCIS on a well in a 24-well plate (cell imaging plate from Eppendorf, Hamburg, Germany) with (Figure 2a) a schematic of the well plate, (Figure 2b) a photograph of a selection of 3 × 3 series of electrodes from the top of the 24-well plate, (Figure 2c) definitions of the electrode dimensions, (Figure 2d) a photograph of the electrodes from below with a scale bar and (Figure 2e) a cross-sectional schematic of the electrodes. Screen-printed annular electrodes of gold on a polymer film with a pressure sensitive adhesive from the Danish Technological Institute (inner radius of disc, $r_{c}\;$= 2 mm, inner radius of ring, $r_{ai}\;$= 4 mm, and outer radius of ring, $r_{ao}\;$= 7 mm; see Figure 2b) are attached to the backside glass plate of the 24-well plate. A pair of spring-loaded pins (RS-PRO Rounded 2-part Spring probe with a length of 25 mm, a diameter of ø1.36 mm) ensures electrical contact to the annual electrodes. The pins are attached to an N-connector mounted in a test station and connected to a Keysight 4291B Impedance Analyser (Keysight Technologies, Santa Rosa, CA, USA) recording impedance data from 1 MHz to 600 MHz with a voltage amplitude of 1 V. An aluminium lid with attached heater elements for temperature control covers the 24-well plate and shields for electr omagnetic interference. Inside the lid, a calibrated Pt100 sensor (AccuMac Technology, Gilbert, AZ, USA) measured by a Fluke 1529 thermometer (Fluke Corporation, Everett, WA, USA) readout monitors the temperature. The measurements are made with temperature control inside an air-bath [26] with a setpoint just below the target temperature and with separate heaters ensuring temperature stabilization at the target temperature. The data from all instruments are acquired using a software code written in a combination of National Instruments LabWindows/CVI, version 20.0.0 and Visual Studio C++, 2022, version 17.13.5.

The data from the lab-bottle using the configurations in Figure 1d,e and the data from the 24-well plate using the configuration in Figure 2 are recorded at two different frequency ranges. The impedance analyser is most suitable for the well plate configuration, because it is designed for coaxial cell configurations, and in the well plate configuration the signal is guided coaxially to the two pins making contact with the electrodes (see Figure 2d). The frequencies achievable for the combination of the 24-well plate/impedance analyser is in the high range (5–600) MHz. In the lab-bottle configuration, the wires need to be split into two wires, one making contact with the bottom electrode and one making contact with the upper electrode. Stray fields are eliminated in the case by the guard, a feature which cannot be made with the impedance analyser. The LCR-meter is therefore most suitable for the lab-bottle configuration, and as mentioned above, the frequencies in this case are limited to the low frequency range (≤2 MHz), whereas the impedance analyser only operates at higher frequencies (≥5 MHz).

For the data displayed in the present paper, the configurations in Figure 1d,e are used for measurements on UPW and KCl solutions up to 100 mS/m. At higher conductivities, 1 L glass-bottles with KCl solutions are inserted in the flow loop replacing the electrolytic conductivity coaxial DFM primary cell [27] and the liquid control system. The configurations in Figure 2 are used for NCIS measurements on pure water and KCl solutions, where all KCl solutions used as reference materials are traceable to the DFM differential primary cell [28]. For NCIS measurements, bacterial cultures of *S. epidermidis* (ATCC 14990) and *E. coli* (ATCC 2592) were used, Brain Heart Infusion Broth from Sigma-Aldrich, Merck, Darmstadt, Germany (hereafter referred to as BHI-medium) is used as growth medium in a lab-bottle. The bacterial growth measurements were made using the configurations in Figure 1a,e,f and in a well of a 24-well plate using the configuration in Figure 2. Prior to a series of bacterial growth measurements, the lab-bottle comprising the BHI-medium is sterilized in an autoclave, and the 24-well plate is taken out of the sterile package. Comparison of bacterial growth data of *S. epidermidis*, a Gram-positive bacterium, and *E. coli*, a Gram-negative bacterium, between the NCIS measurements and on-line OD measurements at 600 nm is made by collecting samples from the 3-way valve depicted in Figure 1f at time intervals ($t_{sample}$) of 0.5–1 h using an Ultrospec 10 (Biochrom, Cambridge, UK). The BHI-medium is circulated with a flow rate of 85 mL/min. The circulation time of the fluid between the output and input of the pump is <2 s, which is much lower than $t_{sample}$.

Additional samples are collected for off-line bacterial species analysis using Raman spectroscopy assisted by machine learning. The Raman microscope uses an excitation wavelength of 785 nm. A 100× microscope objective is used for focusing the excitation laser (spot size~1 μm), collection of the backscattered Raman light, and visual imaging. Raster scanning is achieved with an automated XYZ stage. The in-build microscope has a field-of-view of approximately 60 μm × 60 μm, and Raman spectra are collected at a wavenumber shift from 700 cm^−1^ to 1600 cm^−1^ by a Horiba spectrometer HR320 (HORIBA UK, Northampton, UK). A convolutional neural network (CNN) is used on hyper-spectral data maps with multiple Raman spectra of bacteria as developed in the work by Ho et al. [29]. Prior to feeding Raman spectra into a CNN, preprocessing is necessary to enhance signal quality and reduce noise. The spectra have been baseline-corrected, normalized, and noise peaks have been removed. The model contains 17 classes of bacteria and 1 non-bacterial (calcium fluoride (CaF_2_) background) and each class contains 961 Raman spectra. The CNN processes Raman spectral data through layers that detect features, reduce complexity, and make predictions, resulting in an efficient analysis of the spectral information. Therefore, the CNN effectively analyses the Raman spectra of bacteria through machine learning of spectral patterns, enhancing the detection accuracy and automating the identification process. The training accuracy is 99% using a training/validation split of 90%/10%. The details of the homebuilt Raman microscope and the machine learning algorithm can be found in Refs. [30,31]. The Raman data sets for machine learning are described in Appendix A.

## 3. Results and Discussion

### 3.1. Measurements on Pure Water and KCl Solutions

Figure 3 presents plots of changes in NCSI data as a function of conductivity of KCl solutions. The data in Figure 3a–d have been recorded by the Keysight E4980 LCR meter from a 500 mL glass lab-bottle as illustrated schematically in Figure 1b,e using seven different solutions with conductivities ranging from 5.5 µS/m (UPW) up to 1.7 S/m. The data in Figure 3e–h have been recorded by the Keysight 4291B Impedance Analyser from a well of a 24-well plate as illustrated schematically in Figure 2, where the well was filled with either pure water saturated with ambient CO_2_ with a conductivity of ~0.1 mS/m or filled with KCl solutions at conductivities ranging from 10 mS/m to 30 S/m. In both cases, the temperature was kept at 25.00 °C with a stability better than 0.02 °C.

The effects of the insulating glass walls of the lab-bottle or the insulating glass plate of the 24-well plate are eliminated by plotting the impedance data in Figure 3 as changes, subtracting from the impedance data the values for UPW/pure water acting as a baseline solution. It should be noted that in Figure 3 and all subsequent plots comprising impedance data, the plots are presented as impedance changes, and depending on the baseline, the data can be negative or positive. In Figure 3, data of real parts of the impedance change are plotted as blue diamond symbols, and data of imaginary parts of the impedance change are plotted as reddish-brown square symbols. A possible drift in pure water caused by the ambient conditions has a negligible or minor effect on the data set, since the conductivities of the other solutions are at least two orders of magnitude larger than the conductivity of pure water.

For each of the two data sets recorded by the Keysight E4980 LCR meter and the Keysight 4291B Impedance Analyser as illustrated schematically in Figure 1d,e and Figure 2, respectively, plots are displayed for two sets of different frequencies as indicated: (a) 10 kHz, (b) 50 kHz, (c) 300 kHz and (d) 1000 kHz for the lab-bottle configuration, and (e) 5 MHz, (f) 30 MHz, (g) 100 MHz and (h) 300 MHz for the 24-well plate configuration. The corresponding standard deviations of the real part of the impedance changes and the imaginary part of the impedance changes are approximately 0.7% and 0.04%, respectively, for the data sets in Figure 3a–d, 3.6% and 0.14%, respectively, for the data sets in Figure 3e, and 0.7% and 0.06%, respectively, for the data sets in Figure 3f–h.

The blue solid curves and reddish-brown curves in Figure 3 are plots of computations based on an expression as derived in the following from the equivalent circuit diagram in Figure 4. Since the capacitance ($C_{g}$) related to the insulating glass wall or glass plate is a constant, it is eliminated in the computation of changes in the impedance. In this case, the equivalent circuit diagram is reduced to the simplest possible diagram for an electrolytic conductivity cell being a parallel RC-circuit [32], and the real part of the impedance change can be written as follows:(1)$$Re\left\{\Delta Z\right\}=\frac{R_{0}}{1+{\left(\omega R_{0}C_{1}\right)}^{2}}-\frac{R_{b}}{1+{\left(\omega R_{b}C_{1}\right)}^{2}}$$
where ω=2πf is the angular frequency with f being the frequency, and where the bulk solution resistance of the actual solution ($R_{0}$) and the baseline solution ($R_{b}$) is given by the following:(2)$$R_{i}=\frac{K_{cell}}{\kappa_{i}},\;i=0,\;b$$
where $\kappa_{i}$ is the conductivity of the solution, $K_{cell}$ is a cell constant related to the physical dimensions of the electrodes, and the capacitance ($C_{1}$) is related to $K_{cell}$, as follows:(3)$$C_{1}=\frac{\varepsilon_{r}\varepsilon_{0}}{K_{cell}}$$
$\varepsilon_{r}$ being the relative permittivity (set to 78 for water) and $\varepsilon_{0}$ being the vacuum permittivity. Using Equations (2) and (3), Equation (1) can be written as follows:(4)$$Re\left\{\Delta Z\right\}=\frac{\varepsilon_{0}\varepsilon_{r}}{C_{1}}\left(\;\frac{\kappa_{0}}{\kappa_{0}^{2}+{\left({\omega \varepsilon }_{r}\varepsilon_{0}\right)}^{2}}-\frac{\kappa_{b}}{\kappa_{b}^{2}+{\left({\omega \varepsilon }_{r}\varepsilon_{0}\right)}^{2}}\right)$$

Similarly to Equation (4), (the negative of) the imaginary part of the impedance change (reddish-brown curves in Figure 3) can be written as follows:(5)$$-Im\left\{\Delta Z\right\}=\frac{{{\omega (\varepsilon }_{r}\varepsilon_{0})}^{2}}{C_{1}}\left(\frac{1}{\kappa_{0}^{2}+{\left(\omega \varepsilon_{r}\varepsilon_{0}\right)}^{2}}-\frac{1}{\kappa_{b}^{2}+{\left(\omega \varepsilon_{r}\varepsilon_{0}\right)}^{2}}\right)$$

All blue and reddish-brown curves in Figure 3a–d have been plotted using Equations (4) and (5) with $C_{1}$ = 6.0 pF, and all blue and reddish-brown curves in Figure 3e–h have been plotted using Equations (4) and (5) with $C_{1}$ = 2.0 pF. This implies that each of the two sets of four blue curves and four reddish-brown curves in Figure 3 is derived using only one adjustable parameter ($C_{1}$).

As observed from the data in Figure 3, there is a maximal value of $Re\left\{\Delta Z\right\}$ at each frequency. According to Equation (4), it occurs at the following conductivity:(6)$$\kappa_{0,d}=\omega \varepsilon_{r}\varepsilon_{0}=2\pi f\varepsilon_{r}\varepsilon_{0},$$
which only depends on the permittivity of the solution and the frequency. The maximal values are, for example, $\kappa_{0,d}$ = 4.3 × 10^−5^ S/m at f = 10 kHz, $\kappa_{0,d}$ = 4.3 × 10^−3^ S/m at f = 1000 kHz and $\kappa_{0,d}$ = 1.3 S/m at f = 300 MHz, matching the data in Figure 3. The data show that there is a dynamic range of $\kappa_{i}$ of about one order of magnitude below $\kappa_{0,d}$ and one order of magnitude above $\kappa_{0,d}$ with optimal sensitivity, where the slopes of the curves for $Re\left\{\Delta Z\right\}$ are large; the dynamic range of $-Im\left\{\Delta Z\right\}$ is smaller, though. Apart from the cases of low frequency and high conductivity in the lab-bottle configuration (see Figure 3a,b), the simple model describes the data set well. The model works well even up to a KCl solution of 30 S/m corresponding to $R_{0}$ = 3.8 Ω for the electrode configuration in Figure 2. The fact that each data set at a separate value of conductivity in Figure 3 has been recorded at a separate series of measurements confirms the reproducibility of the NCSI measurements.

More elaborate circuit models involving one or more constant phase elements and an inductor were tested for eligibility. Such models were used in the past for modelling contact impedance spectroscopy data on KCl solutions (see e.g., Ref. [32]). A useful model should reflect physical parameters. For the present set of NCSI data, our model, despite its simplicity, effectively captures all significant physical aspects while providing a robust fit to the data, with no alternative model offering a superior balance between accuracy and physical relevance.

The plots in Figure 3 of impedance data as a function of conductivity of the KCl solutions are like other plots of mechanical, electrical or electromechanical dissipations, including plots of real and imaginary parts of permittivity or impedance magnitude and phase as a function of frequency [33,34]. This is a direct reflection of the Drude model [35,36].

### 3.2. Measurements of Growth of Bacteria

As mentioned in Section 2, for the present bacterial growth measurements, BHI was used as a growth medium. Using an in-house built coaxial cell comprising electrodes of stainless steel directly in contact with the BHI-medium and with a cell constant of $K_{cell}$ = (10.6 ± 0.4) m^−1^, the plain BHI-medium was measured by the Keysight E4980 LCR meter to have conductivity of 1.49 S/m at 25 °C and 1.75 S/m at 37 °C, corresponding to a temperature coefficient of approximately 2%/°C, which is similar to the temperature coefficient of KCl solutions [26].

NCIS data are plotted in Figure 5 for growth of *S. epidermidis* using two different measurement configurations: (Figure 5a–c) for a lab-bottle recorded by the Keysight E4980 LCR meter at 50 kHz (configuration in Figure 1d) from a time, t= 4.5 h to 7.5 h after inoculation of the bacterial culture; and (Figure 5d–f) for a well in a 24-well plate recorded by the Keysight 4291B Impedance Analyser at 160 MHz (configuration in Figure 2) from t= 12 h to 22 h after inoculation of the bacterial culture. NCIS data recorded at (Figure 5a–c) t= 4.5 h and (Figure 5d–f) t= 12 h, respectively, are subtracted from the data in each plot as baselines. In Figure 5a, $-Im\left\{\Delta Z\right\}$ is plotted as a function of $Re\left\{\Delta Z\right\}$ at *t* = 6.0 h (violet square symbols), *t* = 7.0 h (green triangular symbols) and *t* = 7.5 h (black circular symbols), and in Figure 5d at *t* = 15.5 h (violet square symbols), 17.5 h (green triangular symbols) and *t* = 22 h (black circular symbols), where the rightmost arrows indicate the direction of the impedance data for increasing frequency. These plots are of a type commonly referred to as “Nyquist diagrams” [1]. The blue symbols in Figure 5b,e are plots of $Re\left\{\Delta Z\right\}$ as a function of t, and the reddish-brown symbols in Figure 5c,f are plots of $-Im\left\{\Delta Z\right\}$ as a function of t. The average and standard deviation of temperature during the bacterial growth is (37.026 ± 0.011) °C for the data in Figure 5a–c and (37.002 ± 0.004) °C for the data in Figure 5d–f. With a temperature coefficient of 2%/°C, temperature effects on the data are small and can be neglected. Furthermore, the overall stability of the system was tested with continuous real-time measurements of pure BHI media, shown in grey dotted plots in Figure 5b,c,e,f, demonstrating stability and robustness over the entire course of the experiments done on different days. The plots show minimal to no effect on measurements.

According to common bacterial growth patterns [37], each plot in Figure 5b,c,e,f includes a lag phase prior to the onset of the exponential growth, a log phase exhibiting exponential growth and a stationary phase, where the exponential growth is completed, but prior to the phase of cell deaths. The impedance data at the corresponding frequencies (50 kHz and 160 MHz) are indicated by the leftmost arrows in Figure 5a and Figure 5d, respectively. At these frequencies, a positive change of $Re\left\{\Delta Z\right\}$ is observed in Figure 5b and a negative change of $Re\left\{\Delta Z\right\}$ is observed in Figure 5e. For $-Im\left\{\Delta Z\right\}$, a positive change is observed both in Figure 5c,f. According to the literature, the impedance spectrum of a biological cell depends on the growth medium and the cell [12,17,33]. Bacterial cells can be modelled as a series of small *RC* circuits [10,12] and the impedance of the medium for bacterial growth depends on the concentration of cells [18,38], as observed in Figure 5. At several kilohertz and up to the megahertz range, the cell shape and size dominate the impedance, and this is referred to as Maxwell–Wagner polarization [5,12,33]. The data in Figure 5a–c have been recorded within this frequency range. At frequencies of several megahertz, the electric field penetrates the cell membrane, and the impedance is dominated by the cell cytoplasm properties [39]. The data in Figure 5d–f have been recorded within this frequency range. The black dashed curves in Figure 5b–f are data fitting using the following expression for a sigmoidal function as a function of time (t):(7)$$U\left(t\right)=\frac{U_{a}}{1+e^{-\left(t-t_{0}\right)/\tau }}$$
where τ is a volume bacterial growth time constant, and $t_{0}$ is a time lag. For the data-fitted curves in Figure 5b, the values of $U_{a}$, τ and $t_{0}$ are for $Re\left\{\Delta Z\right\}$: $U_{a}$ = 4550 Ω, τ = 0.25 h and $t_{0}$ = 6.0 h; in Figure 5c for $-Im\left\{\Delta Z\right\}$: $U_{a}$ = 7500 Ω, τ = 0.25 h, and $t_{0}$ = 6.3 h; in Figure 5e for $Re\left\{\Delta Z\right\}$: $U_{a}$ = −10.0 Ω, τ = 1.0 h, $t_{0}$ = 17.2 h; and in Figure 5f for $-Im\left\{\Delta Z\right\}$: $U_{a}$ = 6.0 Ω, τ = 1.0 h, and $t_{0}$ = 18.0 h.

It is observed that the data can be modelled by sigmoidal functions, which is an approach commonly used in modelling the growth phase of bacteria in liquid culture [37]. The time lag and growth time constant are smaller for bacterial growth in the lab-bottle than in the well of the 24-well plate. This is expected, since the growth of *S. epidermidis* and *E. coli* in the lab-bottle were made with stirring (400 rpm), whilst there was no stirring in the case of growth in the well.

Using the measurement configuration as illustrated schematically in Figure 1e with the lab-bottle inserted in the flow-loop system in Figure 1c, comparisons between NCIS data for growth of *S. epidermidis* and *E. coli* bacterial cultures in BHI-medium in a lab-bottle and on-line OD measurements at 600 nm have been made, and the results are depicted in Figure 6. The onset of bacterial growth occurs approximately four hours after inoculation with *S. epidermidis*, approximately two hours after inoculation with *E. coli*, and up to ten hours after inoculation, the stability phase in the growth process is reached. Each data set recorded at this point of time acts as a respective baseline for the corresponding data of the real part of the impedance changes ($Re\left\{\Delta Z\right\}$) and the corresponding changes in the optical reflection (∆OR), as plotted in Figure 6. For the data sets in Figure 6, the average and standard deviation of temperature during the bacterial growth is (37.000 ± 0.006) °C for *S. epidermidis* and (36.998 ± 0.019) °C for *E. coli*. To facilitate the comparison of the data sets, normalized responses computed from the data are plotted in Figure 6a,e as a function of time for ∆OR at 800 nm (open circular reddish-brown symbols), $Re\left\{\Delta Z\right\}$ at 15 kHz (open square blue symbols) and at 300 kHz (filled square blue symbols), and OD at 600 nm (filled circular reddish-brown symbols). For each data set (u(t)), computation of the normalized response of the data set ($u_{N}(t)$) in Figure 6 is made according to $u_{N}\left(t\right)=\left(u\left(t\right)-u_{min}\right)/\left(u_{max}-u_{min}\right)$, where $u_{min}$ is the minimum value and $u_{max}$ is the maximum value in the data set. In Figure 6, $u_{N}(t)$ is plotted for OD and $Re\left\{\Delta Z\right\}$ at 15 kHz and 300 kHz, respectively; but since the response from OR decreases during bacterial growth, (1 − $u_{N}\left(t\right)$) is plotted for ∆OR in Figure 6. Individual data fitting is included in Figure 6a,e with plots based on Equation (7) for $Re\left\{\Delta Z\right\}$ at 15 kHz and 300 kHz (leftmost and rightmost dashed blue curves, respectively) and for ΔOR and OD (leftmost and rightmost dashed reddish-brown curves, respectively); the values used for τ and $t_{0}$ are listed in Table 1. For *S. epidermidis*, the time lags ($t_{0}$) correspond to a delay relative to OD of $t_{delay}$ = −1.6 h for $Re\left\{\Delta Z\right\}$ at 15 kHz, $t_{delay}$ = 0 h for $Re\left\{\Delta Z\right\}$ at 300 kHz and $t_{delay}$ = −2.1 h for ∆OR, respectively. For *E. coli*, $t_{0}$ corresponds to a delay relative to OD of $t_{delay}$ = −2.7 h for $Re\left\{\Delta Z\right\}$ at 15 kHz, $t_{delay}$ = −1.0 h for $Re\left\{\Delta Z\right\}$ at 300 kHz and $t_{delay}$ = −2.4 h for ∆OR, respectively.

The same values of $t_{delay}$ are used in the plots as a function of OD in Figure 6b–d,f–h; for (b), (f) $Re\left\{\Delta Z\right\}$ at 15 kHz, (c), (g) $Re\left\{\Delta Z\right\}$ at 300 kHz, and (d), (h) ∆OR. The data plotted as filled circular symbols and the corresponding error bars are average and standard deviations, respectively, of $Re\left\{\Delta Z\right\}$ computed in time slots from half the sampling period ($t_{sample}/2$) before to $t_{sample}/2$ after the time when the corresponding OD data is recorded. The standard deviations are affected by the sampling process, which causes disturbance in the flow and stirring of the BHI-medium. Normally, OD is only applicable in the range from 0.1 to 1.0 [40], and the dashed lines in Figure 6b–d,f–h are linear regressions on the data set for the OD range from 0.16 to 1.3. Using the values of $t_{delay}$ as mentioned above, for the data of *S. epidermidis* in Figure 6b–d the coefficient of determination is (b) R^2^ = 0.9991 for $Re\left\{\Delta Z\right\}$ at 15 kHz, (c) R^2^ = 0.9965 for $Re\left\{\Delta Z\right\}$ at 300 kHz, and (d) R^2^ = 0.9953 for ∆OR. For the data of *E. coli* in Figure 6f–h, the coefficient of determination is (f) R^2^ = 0.9912 for $Re\left\{\Delta Z\right\}$ at 15 kHz, (g) R^2^ = 0.9947 for $Re\left\{\Delta Z\right\}$ at 300 kHz, and (h) R^2^ = 0.9984 for ∆OR. At OD > 1.3, the data for $Re\left\{\Delta Z\right\}$ at 15 kHz (see Figure 6b,f) and the data for ∆OR (see Figure 6d,h) deviates from the dashed lines with smaller rate of change and tends to saturate. As observed, this behaviour is similar for the data of *S. epidermidis* and *E. coli.* The data for $Re\left\{\Delta Z\right\}$ at 300 kHz has different behaviour for *S. epidermidis* and *E. coli* (compare Figure 6c with Figure 6g). For *S. epidermidis*, the data deviate with a larger rate of change for OD > 1.3, showing that these data saturate at a higher level than the OD data and that NCIS exhibits a larger dynamic range than OR and OD. On the other hand, for *E. coli*, the data deviate with a negative rate of change for OD > 1.3, indicating that the *E. coli* bacteria have entered into the mortality phase [37] in this case.

It is observed in Figure 6a and from the row for $t_{0}$ in Table 1 that there is a relatively small difference in $t_{0}$ of 0.45 h for *S. epidermidis* and −0.3 h for *E. coli* for the growth curves between $Re\left\{\Delta Z\right\}$ at 15 kHz and ∆OR, and that there is essentially no difference in $t_{0}$ between the growth curves of $Re\left\{\Delta Z\right\}$ at 300 kHz and OD, and for *E. coli* the difference in $t_{0}$ is only 1.0 h. The observation suggests that the data for both $Re\left\{\Delta Z\right\}$ at 15 kHz and ∆OR primarily originates from bacterial growth in the vicinity of the inner lab-bottle wall. For ∆OR this is expected, because the light is illuminated and collected from an area close to the inner lab-bottle wall.

Contrarily, the data for OD and $Re\left\{\Delta Z\right\}$ at 300 kHz both originate from bacterial growth in the bulk of the medium, because the PTFE tubes are positioned close to the centre of the lab-bottle, where samples for the OD measurements are collected, and there is essentially no or minimal time delay between the data for $Re\left\{\Delta Z\right\}$ at 300 kHz and OD. This is also supported by a unified model, which will be derived below and where the time lag for OD and the time lag for $Re\left\{\Delta Z\right\}$ for growth in the bulk are set to the same value (see Table 1 and Figure 7). Here, it is also important to note that for both bacterial cultures, *S. epidermidis* and *E. coli* growth time constant (τ) for $Re\left\{\Delta Z\right\}$ at 300 kHz is larger (τ = 1.0 h) compared to τ for the three other data sets (τ = 0.5 h) (see Table 1), and this can be explained by the unified model in Figure 7.

Both $Re\left\{\Delta Z\right\}$ data sets at 15 kHz and 300 kHz are within the range of Maxwell–Wagner polarization, where cell shape and size dominate the impedance [5,12,33], and the differences in $t_{0}$ and τ from the individual data fitting to the two data sets are puzzling. However, one major difference between the NCIS at 15 kHz and at 300 kHz is the fact that the impedance level at 15 kHz (hundreds of kiloohms) is much higher than at 300 kHz (a few tens of kiloohms). The inhomogeneous bacterial growth, which is confirmed by the difference in the time lags of the OD and ∆OR data of $t_{delay}$ = −2.1 h for *S. epidermidis* and $t_{delay}$ = −2.4 h for *E. coli*, therefore affects the measured impedance differently at the different frequencies. The unified model as depicted in Figure 7 illustrates the mechanism and will be explained in the following.

Assume that the conductance of current flowing between the top surface of a solution in the lab-bottle and the bottom of the lab-bottle occurs through an impedance entity (Z) comprising two separate regions supporting bacterial growth and with different resistivities; a central rod representing the bulk solution of the bottle comprising a radius $\left(r_{1}\right)$ and a height (h); and a surrounding cylinder with an inner radius $r_{1}$, an outer radius $r_{2}$ equal to the inner radius of the lab-bottle, and the height (h), where h is determined by the vertical distance between the top and bottom electrodes. From the onset of bacterial growth (see Figure 7), the resistivity indices $\rho_{1}$ and $\rho_{2}$ are defined as measures of change in the resistivity in the central rod and surrounding cylinder, respectively. The cross-sectional areas of the central rod and surrounding cylinder are $\pi r_{1}^{2}$, and $\pi \left(r_{2}^{2}-r_{1}^{2}\right)$, respectively; and the real part of the impedance change in the solution is determined by the change in the two regions in parallel (see Figure 7) and can be written as follows:(8)$$Re\left\{∆Z\right\}=\frac{h\rho_{1}\rho_{2}}{\pi r_{1}^{2}\rho_{2}+\pi \left(r_{2}^{2}-r_{1}^{2}\right)\rho_{1}}\;r_{1}\leq r_{2}$$

The radius $r_{2}$ in Equation (8) is a constant and $r_{1}$ is assumed to grow with a rate of growth from the lab-bottle wall, which is determined by the cubic root of the volume exponential bacterial growth rate:(9)$$r_{1}=r_{2}e^{-\left(t-t_{0w}\right)/(3\tau_{u})},\;t\geq t_{0w},\;r_{1}=r_{2},\;t<t_{0w},$$
where $t_{0w}$ is a time lag and $\tau_{u}$ is the time constant of the volume bacterial growth. For the sake of simplicity, $\rho_{2}(t)$ is set to a constant and $\rho_{1}(t)$ is assumed to vary according to Equation (7),(10)$$\rho_{1}\left(t\right)=\frac{\rho_{a1}}{1+e^{-\left(t-t_{0b}\right)/\tau_{u}}};\;\rho_{2}(t)=\rho_{a2},$$
where $t_{0b}$ is a time lag for the bacterial growth in the bulk medium being in the vicinity of the centre of the lab-bottle.

Using Equations (8)–(10), an expression for the bacterial growth ($U_{u}\left(t\right)=Re\{\Delta Z\left(t\right)\}U_{n}\pi r_{2}^{2}/(\rho_{a1}h)$) can be written,(11)$$U_{u}\left(t\right)=\frac{U_{n}}{e^{-2\left(t-t_{0w}\right)/(3\tau_{u})}\left(1+e^{-\left(t-t_{0b}\right)/\tau_{u}}\right)+\left(1-e^{-2\left(t-t_{0w}\right)/(3\tau_{u})}\right)\frac{\rho_{a1}}{\rho_{a2}}},\;t\geq t_{0w}\;U_{u}\left(t\right)=\frac{U_{n}}{1+e^{-\left(t-t_{0b}\right)/\tau_{u}}},\;t<t_{0w},$$
where $U_{n}$ is a normalization constant.

The leftmost and rightmost solid black curves in Figure 6a,e are plots based on Equation (11) for $Re\left\{\Delta Z\right\}$ at 15 kHz and at 300 kHz, respectively. In Equation (11), the lag time ($t_{0w}$) for bacterial growth from the walls has been set to the value corresponding to the individual data fitting for $Re\left\{\Delta Z\right\}$ at 15 kHz. The lag time ($t_{0b}$) for bulk bacterial growth has been set to the value corresponding to the individual data fitting for $Re\left\{\Delta Z\right\}$ at 300 kHz. The time constant ($\tau_{u}$) has been set to the value corresponding to τ used in the individual data fitting for $Re\left\{\Delta Z\right\}$ at 300 kHz. The values of $\tau_{u}$, $t_{0w}$ and $t_{0b}$ are listed in the last three rows in Table 1 for both bacterial cultures, *S. epidermidis* and *E. coli*. The ratio of the resistivity indices in Equation (11) is set to $\rho_{a1}/\rho_{a2}$ = 3.5 for both bacterial cultures for the computation of the leftmost black solid curve corresponding to a higher bulk resistivity index representing the behaviour of $Re\left\{\Delta Z\right\}$ at 15 kHz, and it is set to $\rho_{a1}/\rho_{a2}$ = 0.7 for *S. epidermidis* and $\rho_{a1}/\rho_{a2}$ = 0.5 for *E. coli* for the computation of the rightmost black solid curve corresponding to a lower bulk resistivity index and representing the behaviour of $Re\left\{\Delta Z\right\}$ at 300 kHz. As observed for the two solid black curves in Figure 6a,e, the unified model based on Equation (11) explains the frequency dependence of $Re\left\{\Delta Z\right\}$ both at 15 kHz (with the data plotted as open square blue symbols), and at 300 kHz (with the data plotted as filled square blue symbols).

It is observed in Figure 6 and from the values of $t_{0w}$ and $t_{0b}$ in Table 1, where $t_{0w}$ = 5.6 h and $t_{0b}$ = 7.2 h for *S. epidermidis*, and $t_{0w}$ = 3.7 h and $t_{0b}$ = 5.4 h for *E. coli*, that *E. coli* initiates growth earlier than *S. epidermidis*. This is in agreement with the literature, where it is reported that *E. coli* is a fast-growing bacterium and *S. epidermidis* is a slower-growing bacterium [12]. The unified model also agrees with comparison data between contact impedance spectroscopy and OD measurements as reported in the literature, including the report by Muñoz-Berbel et al. [41], who monitored bacteria concentrations extracted from an incubator and compared bacteria concentration values from contact impedance spectroscopy in the frequency range from 10 Hz to 100 kHz with OD values at 550 nm. The authors observed earlier detection using impedance spectroscopy than OD, and related the faster response to a better limit of detection of impedance spectroscopy [41]. A similar observation has been made by Park et al. [42], who used an interdigitated and wave-shaped electrode as a capacitance sensor for monitoring growth of the bacterial species *Staphylococcus aureus* (*S. aureus*) in the frequency range from 10 Hz to 1 kHz and compared measurements of impedance magnitude index and capacitance index with OD. At eight hours after starting the measurements, their bacterial growth data show that the growth curves of impedance magnitude index and capacitance index have almost reached maximum response, whilst the OD data at 595 nm have a response only about half the maximum value. The present observation in Figure 6 that impedance data at 15 kHz reflecting bacterial growth from a lab-bottle wall precede the OD data with a time shift of $t_{delay}$ = 1.6 h for *S. epidermidis* and $t_{delay}$ = 2.7 h for *E. coli* agree with these reports of contact impedance data compared with OD data.

Apart from inhomogeneous growth behaviour as observed in Figure 6, NCIS provides us with considerably more information compared to standard optical measurements such as OD or OR measuring bacterial concentration. In general, impedance spectroscopy enables analyses of each ionic conduction process including ionic charge transfer dynamics, (ionic and molecular) diffusion processes, cell membrane properties including cytoplasm conductivity and geometric capacitance and resistance of the cell membrane [10,17,38], and it enables analyses of the bacterial metabolism producing substances that change the growth medium impedance [8,9,12,38]. Figure 3 illustrates that different ranges of conductive processes can be monitored at different frequencies, and Figure 5 illustrates the different behaviour of the NCIS spectra for the lab-bottle and 24-plate well at two different frequencies.

For the present measurements, samples of BHI-medium containing bacteria were collected by the end of the growth phase. The samples were dried, positioned on a CaF_2_ microscope slide and analysed by Raman microscopy assisted by machine learning for identification. The prediction maps, as shown in Figure 8, consist of hyperspectral Raman maps with a size of 40 μm × 40 μm for Figure 8a and 30 µm × 30 μm for Figure 8d, and they consist of 1678 and 944 Raman spectra, respectively, in the range from 700 cm^−1^ to 1600 cm^−1^ with 1 μm spacing between the points. The Raman spectra are acquired with five-times averaging and with 3 s integration, and the results from a sample are presented in Figure 8 with (a) and (d) optical images of the scan area, (b) and (e) bacterial species identification with *S. epidermidis* and *E. coli* species, respectively mapped out, and (c) and (f) prediction maps for the rest of the 16 bacterial classes. The overall density surface coverage is 8.0% for *S. epidermidis*, 2.1% for the rest of the bacterial species identification comprising 16 different classes of bacteria, and 89.9% for the CaF_2_ background. Specifically, the model classifies 1551 Raman traces (points) as CaF_2_ background, 118 points as *S. epidermidis*, five points as *Acinetobacter*, four points as *Corynebacterium afermentans* and three points as *S. aureus*. Thus, it is observed that mainly *S. epidermidis* species are identified, which verifies that the data as plotted in Figure 6 originate from growth of *S. epidermidis*. Similarly, the overall density surface coverage of *E. coli* is 99%, showing a crowded surface almost completely occupied by *E. coli*, 0.5% *S. epidermidis* and the remainder comprising 16 other species, confirming that the species used in the NCIS measurements were not contaminated.

### 3.3. Future Directions

In the measurements from a well of a 24-well plate (see configuration in Figure 2) as reported in the present paper, the spring-loaded pins act as contacts to screen-printed annular electrodes of gold on a polymer film with the adhesive being attached to the backside of the 24-well plate. In a more practical design, the spring-loaded pins should preferably have the same shape as the annular electrodes and make direct contact with the backside of the 24-well plate. A practical design will also involve an array of spring-loaded electrodes matching the number of wells of the well plate used. A larger array of pins matching ≥96-well plates would be desirable. The electrodes may be integrated with switching and measurement electronics to accommodate different forms of labware and to facilitate signal processing, signal integrity and data processing speed.

Apart from being a tool in microbiology laboratories working with areas such as antibiotic susceptibility testing or starter cultures for fermentation, the authors expect that NCIS for well plates would be suitable for various applications including monitoring biofilm growth over the passage of time [9]. NCIS may also have applications for analyses on the contents in closed bottles, which must not be exposed to the environment. This includes cultures of micro-organisms with specific requirements such as obligate anaerobes that do not tolerate atmospheric oxygen [43], or measurements of the degradation of reference materials like KCl solutions [32] for shelf-life measurements. Currently, measurements of volatile solutions such as biofuels exhibiting low electrolytic conductivity [44] are challenging, because the solutions change when exposed to atmospheric air. In this case, it will be advantageous to make measurements in sealed containers, which could be fulfilled with properly designed electrodes combined with NCIS.

## 4. Conclusions

Non-contact impedance spectroscopy (NCIS) of bacterial growth in standard laboratory labware and 24-well plates has been reported. The electrodes were applied to the outside insulating walls of 500 mL lab-bottles and to wells of 24-well plates, and they were used for impedance spectroscopy of KCl solutions with conductivities from 5.5 µS/m up to 30 S/m covering six to seven orders of magnitude of conductivity. Except for high conductivities and frequencies for the lab-bottle configuration, the NCIS data could be modelled well using a parallel RC-circuit, which is the simplest possible equivalent circuit diagram for an electrolytic conductivity cell that contains only one adjustable parameter, the capacitance.

Using the presently developed electrodes for NCIS, bacterial growth monitoring of *S. epidermidis* and *E. coli* in BHI-media was demonstrated in 500 mL lab-bottles and in wells of 24-well plates. The real and imaginary parts of the bacterial-induced changes in the growth medium exhibited exponential growth behaviour, as commonly observed for bacteria. For the lab-bottle configuration, a comparison between simultaneously recorded NCIS data, optical reflection (OR) and optical density (OD) was made for growth of two different classes of bacterial cultures, i.e., *S. epidermidis*, a Gram-positive bacterium and *E. coli*, a Gram-negative bacterium. In the OD range from 0.16 to 1.3, good agreement was demonstrated between the NCIS data and OD data with coefficient of determination >0.99. At 300 kHz, the onset of bacterial growth as observed in the NCIS data and OD data was observed to occur with a smaller time difference, which could be set to zero for the *S. epidermidis* data and to 1 h for the *E. coli* data in the analysis. In the case of 15 kHz, the onset of bacterial growth for the NCIS data was observed to occur about 0.5 h for *S. epidermidis* and −0.3 h for *E. coli* later than the OR data, and 1.6 h for *S. epidermidis* and 2.7 h for *E. coli* earlier than the OD data. A unified growth model based on a combination of bulk bacterial growth and bacterial growth from the inner lab-bottle walls towards the centre of the lab-bottle could explain these observations. The observed faster growth of *E. coli* compared to *S. epidermidis* agrees with the literature, where it is reported that *E. coli* is a fast-growing bacterium and *S. epidermidis* is slower in comparison.

Compared to OR and OD, the advantage of NCIS is that it covers growth processes both in the bulk of a container and in the vicinity of the container walls, covering a longer time of the bacterial growth process, a wider dynamic range, and providing more details about the bacterial cell properties. For ease of comparison between NCIS and the state of the art, a table has been added to the Appendix A.

The bacterial species were verified by Raman spectroscopy assisted by machine learning analyses of samples collected by the end of the growth phase. The analyses of two different samples showed that the *S. epidermidis* and *E. coli* species were successfully identified, verifying that the data from NCIS, OR and OD as reported originates from growth of the target species and not contamination. The results indicate that NCIS could be a useful tool for microbiology analyses made on samples in off-the-shelf labware and various types of non-invasive chemical or biochemical analyses on the contents in closed containers.

## Figures and Tables

**Figure 1 sensors-25-02427-f001:**
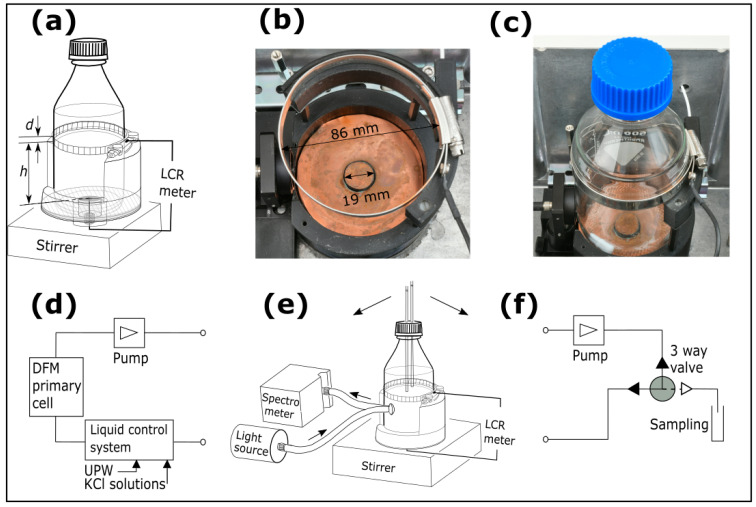
Electrode configuration for non-contact impedance spectroscopy (NCIS) on a 500 mL lab-bottle and measurement system: (**a**) schematics of lab-bottle with a closed lid in a bottle mount, (**b**) a photograph of the electrodes with scale bars, (**c**) a photograph of the electrodes attached to a lab-bottle, (**e**) lab-bottle in the bottle mount, inserted in a flow-loop ((**d**) or (**f**)) and with optical reflection spectroscopy from the vicinity of the inner bottle wall. Measurements in the flow-loop are made for the two types of sample series: (**d**) ultra-pure water and KCl solutions and (**f**) BHI-media comprising bacteria, where samples are collected for on-line and off-line analyses.

**Figure 2 sensors-25-02427-f002:**
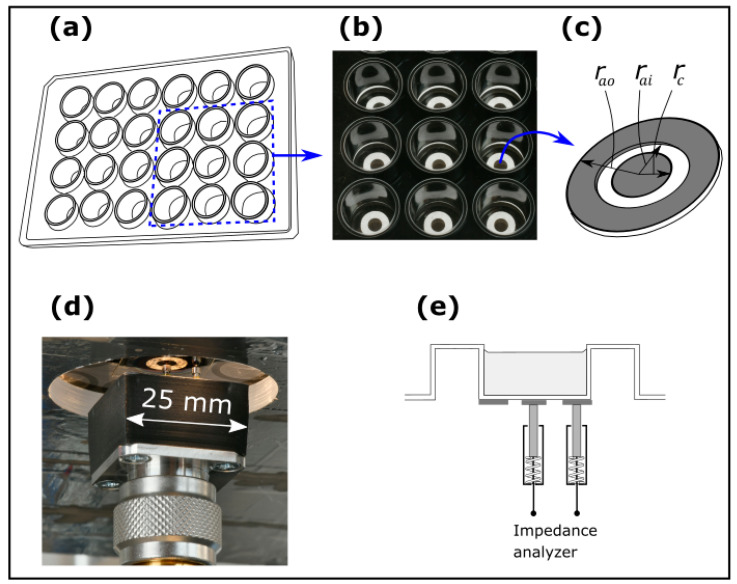
(**a**) Schematic of a 24-well plate, (**b**) a photograph of a selection of 3 × 3 series of electrodes from the top of the 24-well plate, (**c**) definitions of the electrode dimensions for NCIS measurements on a well of a 24-well plate, (**d**) a photograph of the electrodes from below the 24-well plate with a scale bar, and (**e**) cross-sectional schematic of electrodes attached to a well in a 24-well plate.

**Figure 3 sensors-25-02427-f003:**
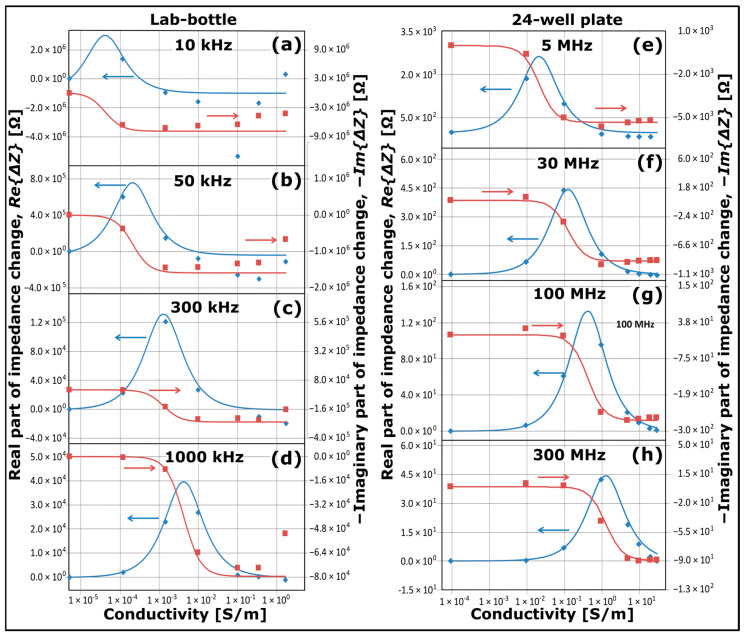
Plots of the real part of impedance change (blue symbols), the imaginary part of impedance change (reddish-brown symbols) as a function of conductivity for KCl solution series in the case of an NCIS measurement configuration comprising (**a**–**d**) lab-bottle and LCR meter, and (**e**–**h**) 24-well plate and impedance analyser. The blue curves and reddish-brown curves are plots using the model as described in the text. The blue and reddish-brown arrows point to the axes for the plots of blue symbols and curves and reddish-brown symbols and curves, respectively.

**Figure 4 sensors-25-02427-f004:**
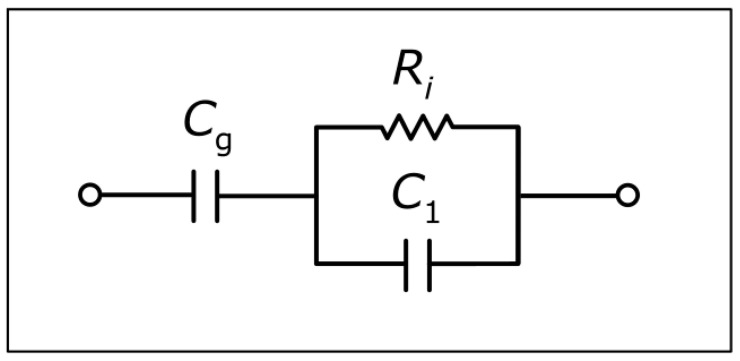
Equivalent circuit diagram for the impedance of a sample as measured by NCIS.

**Figure 5 sensors-25-02427-f005:**
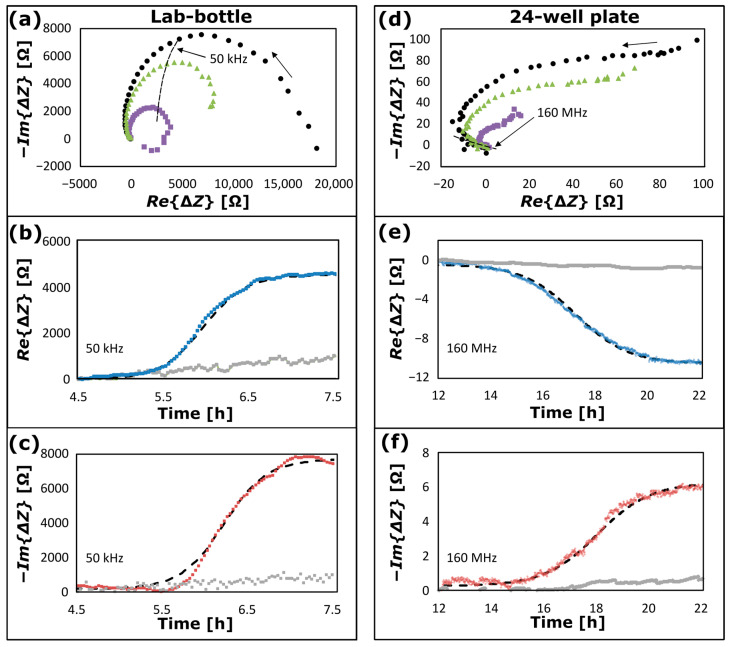
(**a**) NCIS spectra of growth of *S. epidermidis* in a lab-bottle with frequencies from 15 kHz to 2 MHz recorded at *t* = 6.0 h (violet square symbols), *t* = 7.0 h (green triangular symbols) and *t* = 7.5 h (black circular symbols) after inoculation of the bacterial culture; (**b**) the real part of the impedance change (blue curve), and (**c**) (negative of) the imaginary part of the impedance change (reddish-brown curve) as a function of *t* at 50 kHz and corresponding data fitting (dashed curves); (**d**) NCIS spectrum of growth of *S. epidermidis* in a well of a 24-well plate with frequencies from 5 MHz to 600 MHz recorded at *t* = 15.5 h (violet square symbols), 17.5 h (green triangular symbols) and *t* = 22 h (black circular symbols); (**e**) the real part of the impedance change (blue curves), (**f**) (negative of) the imaginary part of the impedance change (reddish-brown curves) as a function of *t* at 160 MHz and corresponding data fitting (dashed curves). The impedance data at 50 kHz and 160 MHz are indicated by the leftmost arrows and dashed curves in (**a**) and (**d**), respectively, and the rightmost arrows indicate the direction of the impedance data for increasing frequency. The grey dotted symbols in (**b**,**c**,**e**,**f**) are corresponding plots of the impedance as a function of time for measurements of pure BHI media with no intentional growth of bacteria.

**Figure 6 sensors-25-02427-f006:**
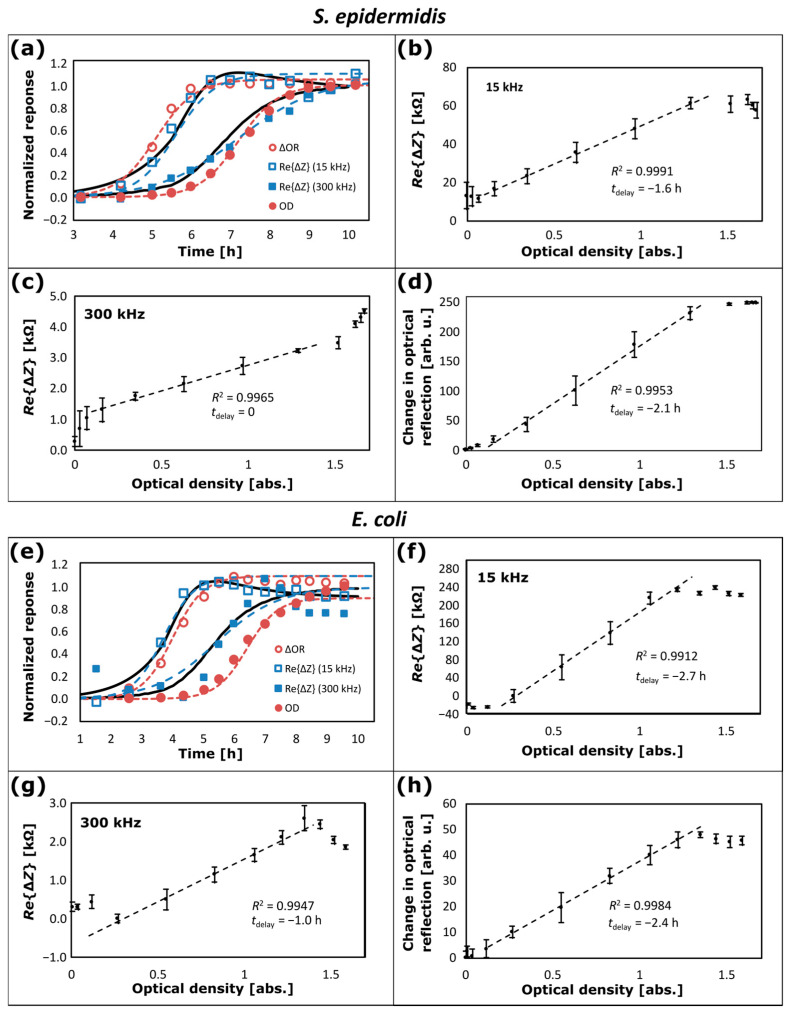
Bacterial growth data for (**a**–**d**) *S. epidermidis* and (**e**–**h**) *E. coli* with (**a**,**e**) normalized response during bacterial growth as a function of time for change in optical reflection (∆OR) (open circular reddish-brown symbols), the real part of the impedance change ($Re\left\{\Delta Z\right\}$) at 15 kHz (open square blue symbols) and at 300 kHz (filled square blue symbols), and optical density (OD) (filled circular reddish-brown symbols); and corresponding $Re\left\{\Delta Z\right\}$ plotted as a function of OD for (**b**,**f**) 15 kHz and (**c**,**g**) 300 kHz; and (**d**,**h**) ∆OR plotted as a function of OD. The dashed lines in (**b**–**d**) and (**f**–**h**) are linear regression lines within the OD range from 0.16 to 1.3. The dashed curves in (**a**,**e**) are data fitting using Equation (7). The leftmost and rightmost solid black curves in (**a**,**e**) are computations based on Equation (11) for 15 kHz and 300 kHz, respectively (see text).

**Figure 7 sensors-25-02427-f007:**
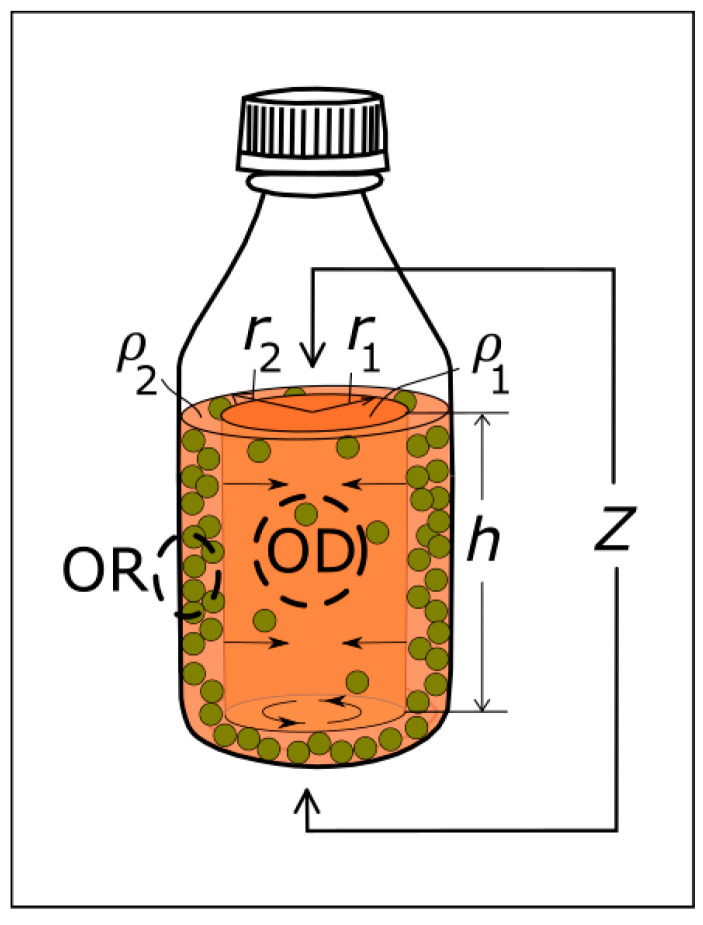
Model of bacterial growth in a lab-bottle, depicting competing growth of bacteria in the medium versus bacteria adhered on inner bottle walls and growing towards the centre of the lab-bottle. The measurement areas are indicated schematically for OR being in the vicinity of the inner lab-bottle wall and for OD being in the vicinity of the centre of the bottle.

**Figure 8 sensors-25-02427-f008:**
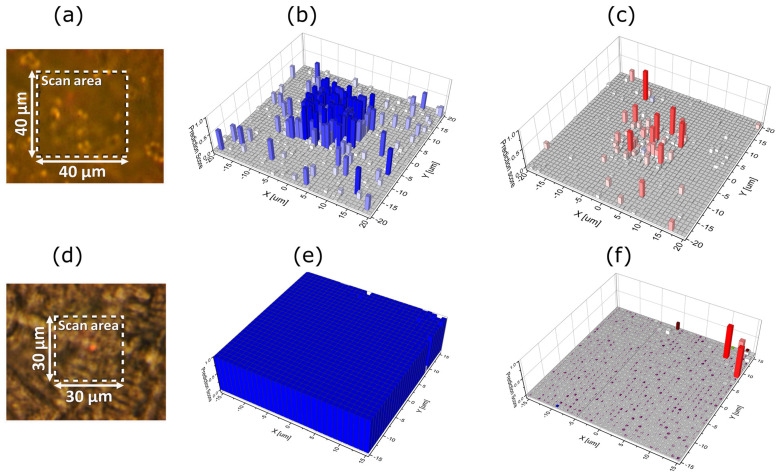
Raman microscopy assisted by machine learning of dried samples and bacterial species identification with (**a**,**d**) showing the optical images of the scan areas for Raman spectroscopy. The prediction maps are shown for (**b**) *S. epidermidis*, (**e**) *E. coli*, (**c**,**f**) all other bacteria comprising 16 different classes.

**Table 1 sensors-25-02427-t001:** Time parameters used for the data fitting in Figure 6.

Time Parameters in Hours	Symbol	OD	Re{ΔZ} 15 kHz	Re{ΔZ} 300 kHz	Change in OR
*S. epidermidis*					
Model based on individual data fitting					
Growth time constant	τ	0.5	0.5	1.0	0.5
Time lag	$t_{0}$	7.2	5.6	7.2	5.1
Unified model as depicted in Figure 7					
Growth time constant	$\tau_{u}$	-	1.0		-
Time lag for growth from the walls	$t_{0w}$	-	5.6		-
Time lag for growth in the bulk	$t_{0b}$	-	7.2		-
*E. coli*					
Model based on individual data fitting					
Growth time constant	τ	0.5	0.5	1.0	0.5
Time lag	$t_{0}$	6.4	3.7	5.4	4.0
Unified model as depicted in Figure 7					
Growth time constant	$\tau_{u}$	-	1.0		-
Time lag for growth from the walls	$t_{0w}$	-	3.7		-
Time lag for growth in the bulk	$t_{0b}$	-	5.4		-

## Data Availability

Data supporting results are available at https://doi.org/10.5281/zenodo.15186864.

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
