# Peer review of "Non-Invasive Real-Time Monitoring of Bacterial Activity by Non-Contact Impedance Spectroscopy for Off-the-Shelf Labware"

_sensors, 2025, doi:10.3390/s25082427_

Round 1
Reviewer 1 Report
Comments and Suggestions for Authors
The manuscript introduces a novel non-contact impedance spectroscopy (NCIS) method with customizable electrodes and demonstrates its practical potential through simplified measurements. While the work is well-structured and technically sound, the following revisions are essential to strengthen its scientific impact and ensure reproducibility:
- The study lacks discussion on potential environmental interference factors (e.g., temperature drift, humidity fluctuations, ambient electromagnetic noise). To address this gap, controlled experiments demonstrating NCIS signal stability under variable conditions should be added. Quantitative validation of measurement reproducibility in unstable environments would significantly strengthen the method’s practical relevance.
- Comparative data from at least two additional bacterial models (ideally representing distinct classifications) should be provided. A discussion linking observed impedance variations to biological features would enhance mechanistic insights.
- Claims of NCIS advantages (e.g., simplicity, accuracy) remain overstated without direct comparisons to established bacterial monitoring methods (e.g., optical density measurements, traditional contact impedance spectroscopy). A tabulated performance comparison should be added, explicitly quantifying sensitivity, detection limits, and operational constraints relative to existing techniques. This would allow readers to contextualize NCIS’s practical value within the field.
- The resolution of physical setup images in Figures 1a and 2c is currently inadequate for readers to discern critical structural or functional details (e.g., electrode spacing, surface morphology). Higher-resolution images with calibrated scale bars, optimized lighting conditions, and explicit dimension labels are required.
Reviewer 2 Report
Comments and Suggestions for Authors
The paper:
Non-invasive Real-Time Monitoring of Bacterial Activity by Non-contact Impedance Spectroscopy for off-the-shelf labware
By Carsten Thirstrup1, Ole Stender Nielsen1, Mikael Lassen1, Thomas Emil Andersen2,3 and Hüsnü Aslan1*
Describes an original experimental configuration to measure the bacterial growth by non-contact impedance spectroscopy, enabling the bacteria in the culture medium do not get in touch with the electrodes. Two set up configuration are examined, one consisting in a glass bottle and the other a well plate system contacted outside the well and both commonly used in biological labs
Overall, the paper describes the feasibility to measure the bacteria density in real time and, as declared, at an early stage, the earliest detection observed in the case of bottle configuration. The configuration is furthermore interesting in view of application without exposure to environment. Also the correlations with OD and OR measurements are intriguing
The paper is well written and suitable for publication in Sensors as it is; however, it would be fine to improve the data acquisition section, to increase the model vs experimental data; I am referring to Figure 3 and 5 where some information (also included in ESI) should b provided
only few questions:
- Why did you choose two frequency ranges one for each configuration for testing the bacteria increase? I am aware that you operate with two different instruments; is there a particular reason for that? Do you want to explore peculiar features of bacteria? Have you some limitation dictated by the experimental configuration (bottle) that does not allow to explore higher frequency ranges?
- It would be nice to have an index of the lowest detectable bacteria concentration if early detection is to be declared;
- as for data representation, NPs for KCl and Water solution or with bacteria in bottle and wells at different times can be advisable, apart the graphs correlating conductivity and impedance components or only points taken at certain times. The hypothesis of a peak is somehow unclear; loss factor peak representation may be more suitable since peak may be correlate to the ionic concentration and probably to the presence of bacteria too.
- How skin effect may affect the measure of the impedance at high frequency; negative values of the -ImZ let to intend some inductive effects for example in Figure 3. Again, providing the whole NP representation, not only a point at a certain frequency may be useful for reader to understand what is behind the results. Also, circuital modelling should take into account the NP behaviors, and sometimes the elementar representation is not enough to extract parameters
- In caption of Figure 3 specify that it refers to KCl solution

Reviewer 3 Report
Comments and Suggestions for Authors
The presented work is interesting, but the article needs improvements, especially regarding the techniques used in the measurements. Figure 1 is difficult to follow and does not clearly show the electrode configuration as stated in the caption title, but several different setups. Things are not clarified in figure 2 either when presenting the 24 well setup. Overall figures are not clearly presented. The Raman spectroscopy identification method is not relevant in this article. Other methods of identification with better sensitivity exist, and also the purpose of the article is not the identification of bacteria but their growth. The authors tested their method only on one type of cells, but for relevance, the experiments should be done with different types of bacteria (shapes and sizes).
Comments on the Quality of English LanguageThe Quality of English Language is overall good.
Round 2
Reviewer 1 Report
Comments and Suggestions for Authors
The revised manuscript has effectively addressed my concerns, and it is now in a form that can be accepted without further modification.
Author Response
Comment 1:
The revised manuscript has effectively addressed my concerns, and it is now in a form that can be accepted without further modification.
Response 1:
We are happy to have effectively addressed the reviewer’s concerns. Thank you for your thoughtful feedback and suggestions.
Reviewer 3 Report
Comments and Suggestions for Authors
The authors have responded to almost all my concerns. However, I still recommend that the section regarding Raman investigations be removed or at least moved to the supplementary material.
Author Response
Comment 1:
The authors have responded to almost all my concerns. However, I still recommend that the section regarding Raman investigations be removed or at least moved to the supplementary material.
Response 1:
We are happy to have successfully addressed almost all Reviewer 3’s concerns.
We associate Reviewer 3’s final recommendation to remove Raman investigations from the manuscript with a concern for relevance to the main topic and possibly to the journal Sensors’ audience.
We acknowledge that the primary focus of our paper is the introduction of a novel, non-invasive method for real-time monitoring of bacterial growth using non-contact impedance spectroscopy. The potential implementation of our method would find place in controlled environments, such as R&D labs, and production lines with strict requirements on contamination prevention and assurance of correct species growth. To this end, Raman spectroscopy with machine learning addresses a limitation of our method, the inability to identify bacterial species and detect contamination. By including this section in the manuscript, we demonstrate the robustness and reliability of our main method.
The audience of Sensors is interdisciplinary, including researchers and professionals from microbiology, biotechnology, and related fields. For these readers, it is essential to see that out method does not introduce contamination and that the bacterial species of interest can be accurately identified. The Raman spectroscopy with machine learning data provides this assurance.
Although there are other methods to identify bacterial species, since this method is also a novel approach developed by our team, we trust that showcasing this innovation is beneficial to the Sensors audience, as it opens avenues for further research and potential integration with other sensing technologies. Moreover, we are actively working on combining the two systems, Non-contact Impedance Spectroscopy with Machine Learning assisted Raman Spectroscopy, which could lead to significant advancements in the field.
From the scientific rigor perspective, including the Raman data strengthens our conclusions and addresses potential criticisms. It shows that we have thoroughly validated our method and considered its limitations.
Given these points, we kindly request that the Raman spectroscopy section remains in the main manuscript. We trust it adds significant value to our paper and is relevant to the Sensors audience.
Thank you once again for your feedback and recommendations.